# Correlation between Choriocapillaris Density and Retinal Sensitivity in Stargardt Disease

**DOI:** 10.3390/jcm8091432

**Published:** 2019-09-10

**Authors:** Rodolfo Mastropasqua, Alfonso Senatore, Luca Di Antonio, Marta Di Nicola, Michele Marchioni, Fabiana Perna, Filippo Amore, Enrico Borrelli, Chiara De Nicola, Paolo Carpineto, Lisa Toto

**Affiliations:** 1Ophthalmology Clinic, University of Marche, 60126 Ancona, Italy; 2Vitreoretinal Unit, Bristol Eye Hospital, University of Bristol, Bristol BS8 1TH, UK; 3Ophthalmology Clinic, Department of Medicine and Science of Ageing, University G. D’Annunzio Chieti-Pescara, via dei Vestini 31, 66100 Chieti, Italy; 4Duke Eye Center, Duke University, Durham, NC 27705, USA; 5Department of Medical, Oral and Biotechnological Sciences, Laboratory of Biostatistics, University “G. d’Annunzio” Chieti-Pescara, via dei Vestini 31, 66100 Chieti, Italy; 6National Center for Services and Research for the Prevention of Blindness and Visual Rehabilitation of the Visually Impaired, 00100 Rome, Italy; 7Ophthalmology Department, San Raffaele University Hospital, 20132 Milan, Italy

**Keywords:** retina, Stargardt disease, OCTA, choriocapillaris, microperimetry

## Abstract

The aim of this work was to characterize the choriocapillaris (CC) in patients with Stargardt disease (STGD) using the swept source widefield optical coherence tomography angiography (SS WF OCTA) and to compare CC perfusion density to retinal sensitivity, analyzed using microperimetry (MP). This cross-sectional study included 9 patients (18 eyes) with STGD and central CC atrophy (stage 3 STGD). The CC was analyzed using SS WF OCTA and areas of different CC impairment were quantified and correlated with retinal sensitivity analyzed using MP. The main outcome measures were the percent perfused choriocapillaris area (PPCA), retinal sensitivity, and correlation between PPCA and retinal sensitivity. Seventeen eyes of 9 patients suffering from stage 3 STGD were analyzed. SS WF OCTA revealed a vascular rarefaction in central atrophic zones and a near atrophy halo of choriocapillaris impairment. In all eyes were noticed a central atrophy (CA) area with absolute absence of CC that corresponded to 0 dB points at MP, a near atrophy (NA) zone of PPCA impairment that included points with decreased sensitivity at MP and a distant from atrophy (DA) zone with higher PPCA and retinal sensitivity values. The mean difference of PPCA and retinal sensitivity between NA and CA and DA and CA was statistical significantly different (*p* < 0.01), the latter showing higher values. A direct relationship between PPCA and retinal sensitivity was found (*p* < 0.001). Choriocapillaris damage evaluated using SS WF OCTA correlates with MP, these data suggest that CC impairment may be a predictor of retinal function in patients with STGD.

## 1. Introduction

Stargardt disease (STGD) is the most common inherited macular dystrophy, with an estimated prevalence of 1:10,000 [1,2]. Advanced stages are characterized by central vision loss with macular atrophy. New insights into pathogenesis have focused Researchers attention on mechanisms of complex lipids synthesis and remodeling, that are thought to be the majors’ determinants of the disease development [3].

Many studies using either fluorescein angiography (FA) and indocyanine green angiography (ICG), or optical coherence tomography (OCT), and more recently OCT angiography (OCTA), have demonstrated choriocapillaris (CC) atrophy in the central atrophic zones of STGD eyes [4,5,6,7,8]. Choriocapillaris atrophy is considered a consequence of retinal pigment epithelium (RPE) atrophy, suggesting a role of RPE in the regulation of CC structure and function [9].

Retinal pigment epithelium is the major responsible of vascular endothelial growth factor (VEGF) production, and VEGF receptors are expressed on the choroidal endothelium facing it. Choroidal vessels provide the vascular support to outer retinal layers suggesting a possible role of CC atrophy in photoreceptors degeneration [4,10]. This also implies that vascular involvement probably plays an important pathogenic role in STGD.

The new swept-source (SS) OCT technologies allow us a better visualization of CC and the higher speed allows the visualization of a wider retinal field of view (wide field, WF) [11].

Furthermore, recent technological improvements have made microperimetry (MP), already commonly used for functional retinal testing, speeder and with a wider range of tested sensitivities [12], being a simpler and more efficient examination of the normal and diseased retina.

The aim of this study was to characterize CC and retinal sensitivity in patients affected by STGD with central CC atrophy, using a SS WF OCTA device and a newer MP device (MP3), and to compare these examinations in order to find correlations between anatomical and functional data that could be of use for clinical practice.

## 2. Methods

### 2.1. Study Participants

In this cross-sectional study, 9 patients (18 eyes) with STGD and central CC atrophy (stage 3 STGD, according to Fishmann classification) were enrolled at the retina center of the Ophthalmology Clinic of University G. D’Annunzio, Chieti-Pescara, Italy. The study was approved by the institutional review board (IRB) and adhered to the tenets of the Declaration of Helsinki. An informed consent, approved by the IRB, was obtained from all patients.

The inclusion criteria were: (1) Clinical, genetic (biallelic mutations in ABCA4 gene) and electrophysiological diagnosis of STGD1 with central CC atrophy and (2) Best Corrected Visual Acuity (BCVA) equal or greater than 1.30 logMAR.

The exclusion criteria were: (1) Any ocular condition apart from STGD capable of reducing visual acuity such as media opacities, glaucoma, macular or diffused retinal pathologies; (2) previous ocular surgery.

The instability of fixation, really common in these patients, wasn’t considered a major limitation, this was due to the presence of an extrafoveal fixation area in all eyes, improved in many patients by previous rehabilitation cycles with MP, and to the efficacy of the new eye trackers present inside the devices used for this study (SS WF OCTA and MP3).

### 2.2. Imaging

#### 2.2.1. OCT and OCTA Imaging

Widefiled angio (12 × 12 mm at least) scans were performed in all eyes using Zeiss PLEX Elite 9000 swept-source OCTA (Carl Zeiss Meditec Inc., Dublin, CA, USA). This device uses a swept source laser with a central wavelength of 1050 nm (1000–1100 nm full bandwidth) and performs at 100,000 A-scans per second, employing a full-width at half-maximum (FWHM) axial resolution of approximately 5 μm in tissue, and a lateral resolution at the retinal surface estimated at approximately 14 μm.

From three scans acquired for each eye was selected the best image, after subjects’ pupils had been dilated with eye drops containing a combination of 10% phenylephrine hydrochloride and 0.5% tropicamide.

In order to calculate central macular thickness, the vertical distance between the inner limiting membrane and the RPE at the foveal center was measured using the inbuilt manual caliper.

#### 2.2.2. Microperimetry

Microperimetry was performed with the microperimeter MP3 by Nidek (Gamagori, Japan). Two examinations were registered for each eye in order to allow two expert observers (LT and AS) to select the best exam, basing their decisions on the number of completed tested points, and false positive and negative mistakes [12]. The automatic eye-tracking function of the MP3 was used during the examinations, which were performed in a dimly lit room. The stimulus was equal in size to a Goldmann III, with a background luminance set at 31.4 apostilbs. The maximum luminance was 10,000 apostilbs, the dynamic range of the stimulus was set at 34 dB, and a single red cross at 1° was employed as a fixation target. 

A 20° area was tested, with the standard “normal fast” protocol that analyzes 33 zones per examination, and each pattern was centered on the preferred fixation area. 

### 2.3. Image Processing

The main outcome measures were (1) the percent perfused choriocapillaris area (PPCA), which represents a measure of the total area of CC perfusion density [13], (2) retinal sensitivity detected with MP3. We selected the best quality OCTA scan, from three obtained in each eye, and one best quality MP3 examination, from two obtained.

To correlate the examinations, we performed a rigid registration of the two selected images, using the “landmark correspondences” plug-in of ImageJ software version 1.50 (National Institutes of Health, Bethesda, MD, USA; http://rsb. info.nih.gov/ij/index.html), and then we considered circular areas (selected as correspondent ROI in ImageJ) of the same size centered on the zones analyzed with MP. Furthermore, we considered four different area sizes on OCTA, with a dimension of 20 × 20, 40 × 40, 50 × 50, and 60 × 60 pixels. Each considered area was then analyzed and correlated to the sensitivity found in the same zone with MP. A total of 33 zones, according to the number of sensitivities tested by MP, were considered for each included eye (Figure 1 and Figure 2).

To evaluate the PPCA and the average signal size, we analyzed the images as already described [13]. In brief, the PPCA was computed as the percentage of pixels in the CC en face image (slab auto-segmented by the instrument) below a “non-perfusion” (or noise) threshold, (National Institutes of Health, Bethesda, MD, USA; http://rsb. info.nih.gov/ij/index.html) as the mean of all the pixel values in the outer avascular retina. The PPCA was thus calculated as the number of pixels falling below the threshold (the total area of the signal voids) divided by the total number of pixels in the analyzed area of CC. Furthermore, the “Analyze Particles” command, which measured and counted all thresholds areas greater or equal to 1 pixel where there was a lack of flow information, furnished us the average size of the signal perfusion.

Moreover, with a topographical method, we defined as outside atrophy (OA) all the zones outside the central atrophy (CA). Within OA zones we identified as near atrophy (NA) all the zones with sensitivity >0 dB, that were the closest to the central atrophic area, evaluated throughout the retinographic image, and that had no other points between them and the atrophic zone. All the other zones outside the atrophy that did not respect these criteria were classified as distant from atrophy (DA). This classification was done by two of the authors (AS and LT) with a complete concordance (An example from a patient in Figure 2).

### 2.4. Statistical Analysis

The minimum required sample size (*n* = 6) was estimated to obtain a percent change in the slope of fixed effect of the linear mixed model equal to 0.3, with at least 80% of desired statistical power level and an alpha error rate of 5%. 

Quantitative variables were reported as mean ± standard deviation while quantitative data were summarized as frequency and percentage. Departures from normal distribution were evaluated for each variable using a Shapiro–Wilk’s test.

Outcomes of interest, retinal sensitivity and PPCA for each area diameters (20 × 20 pixels, 40 × 40 pixels, 50 × 50 pixels, and 60 × 60 pixels), were analyzed using different linear mixed models. Linear mixed models allow explicit modeling of the within- and between-patient variation in the outcome while taking into account that each measurement is not independent from the other since are from both eyes within the same individual. For each patient, a mean difference between values of NA and DA respect to CA zones were evaluated and reported in table as measure of the effect of zone independently of intra-individual variation.

Post-hoc analyses tested for statistically significant difference between NA vs. CA and DA vs. CA. Bonferroni adjustment was applied to post-hoc analyses p-values [14]. 

Moreover, linear mixed models were fitted to test the effect of PPCA at different areas diameters (as independent variables) and retinal sensitivity in the all outside atrophy zones and after stratification in NA and DA zones’ subgroups. The relationship between PPCA and retinal sensitivity was graphically depicted as scatterplot with a linear regression line for all the outside atrophy zones and stratified for NA and DA zones’ subgroup. Significant differences in linear regression coefficient (b) between NA and DA zones were evaluated by Student’s t-test for unpaired data. 

All tests were two-sided, and a level of statistical significance was set at *p* < 0.05. All the statistical analyses were performed using R software environment for statistical computing and graphics version 3.5.2 (R Foundation for Statistical Computing, Vienna, Austria. https://www.R-project.org/)

## 3. Results

### 3.1. Characteristics of Patients Included in the Analysis

Seventeen eyes of 9 patients suffering from stage 3 STGD, according to Fishmann classification, were analyzed. One out of 18 eyes was excluded due to low image resolution. The female/male ratio was 4/5. Mean age was 48.2 ± 15.7 years. The mean BCVA was 1.0 ± 0.1 LogMAR. The mean central macular thickness was 62.00 ± 29.30µm [Table 1; Figure 3].

The mean PPCA OA was 78.1 ± 6.9%, 78.0 ± 5.9%, 78.1 ± 5.4%, and 78.1 ± 5.2% in areas 20 × 20 pixels, 40 × 40 pixels, 50 × 50 pixels, and 60 × 60 pixels, respectively. Instead, the mean PPCA was significantly reduced in CA zones compared to OA zones and was 69.3 ± 9.3%, 68.1 ± 7.0%, 68.1 ± 6.5%, 67.8 ± 5.8% in respectively areas (all *p* < 0.001).

### 3.2. Analysis of CC Density and Retinal Sensitivity

A total of 546 zones corresponding to 33 MP points of a 20° area for each eye of all patients were analyzed, 15 zones of 2 patients (4 eyes) were excluded due to the unreliability of the MP data related to the presence of artifacts.

Optical coherence tomography angiography revealed statistically significant differences of PPCA and sensitivity between CA, NA, and DA zones for all areas (*p* < 0.001). Moreover, post-hoc analyses showed statistically significant PPCA differences between NA vs. CA and DA vs. CA zones (all *p* < 0.001). Mean and SD MP sensitivity was 0 dB in CA zones, 14.0 ± 4.9 dB in NA zones and 25.7 ± 1.3 dB in DA zones (Table 2).

### 3.3. Correlation between CC Density and Retinal Sensitivity

Scatter plot and regression lines [Figure 4] graphically depict the positive relationship between PPCA and retinal sensitivity in each considered area (panels 20 × 20 pixels, 40 × 40 pixels, 50 × 50 pixels and 60 × 60 pixels) for every OA analyzed zone (*p* < 0.001) (Table 3). Similarly, in NA and DA zones a positive relationship was found between PPCA and retinal sensitivity (*p* < 0.001 for each NA area; *p* = 0.002 for 40 × 40 pixels and *p* < 0.001 for 50 × 50 pixels and 60 × 60 pixels DA areas) (Table 3 and Figure 4). In Figure 4, a statistically significant difference between regression coefficient was found in each examined area (all *p* < 0.001).

## 4. Discussion

In this cross-sectional study, we selected patients with STGD in advanced stage (stage 3, according to Fishmann classification) characterized by central CC atrophy that underwent SS WF OCTA and retinal MP assessments.

In our cohort of patients, OCTA revealed in CA zones CC rarefaction with vascular prominence of the underlying Sattler choroidal layer. As expected, all MP points of the CA zone had a 0 dB sensitivity. All the OA analyzed zones showed a higher PPCA than CA and a retinal sensitivity greater than 0 dB; interestingly a positive correlation was found between PPCA and retinal sensitivities. Successively, according to a topographical method we decided to sub-classify the OA in NA and DA; this classification allowed us to perform a sub-analysis of the acquired data. This sub-classification, not previously described in literature, was found to be of really simple execution (as also shown by the total agreement between the two Authors that performed it).

The PPCA is statistically different between NA and DA zones, showing the latter a higher value of PPCA as well as retinal sensitivity was significantly higher in DA compared to NA zones. In addition, a stronger positive correlation was found between PPCA and retinal sensitivity in NA zones if compared to DA zones graphically demonstrated by scatter plots.

Many studies in the literature showed CC rarefaction within the atrophy in patients affected by STGD, this aspect was highlighted as the “dark atrophy” in traditional retinal angiography and then confirmed using fundus autofluorescence (FAF), OCT, en face OCT and, more recently, OCTA by several Authors [7,15,16,17,18,19].

In our previous work the CC analysis was limited to the foveal and parafoveal area and an impairment of the retinal vascular plexuses was also demonstrated; no differentiation between different STGD stages and between atrophic and non-atrophic areas were performed [4].

Outside the atrophy no choroidal flow signal attenuation was noticed by Pellegrini, et al., using a no-SS OCTA technology [16]; instead this aspect was noticed by Muller, et al. [17] that demonstrated an attenuation of the near atrophic areas in STGD patients. Nevertheless, the latter Authors observed lower impairment in STGD patients compared with patients affected by age-related macular degeneration.

In a recent work by Alabduljalil, et al., the attenuation of both photoreceptor and RPE was noticed and their contribution to CC density rarefaction was hypothesized; extra-lesional CC areas were found abnormal and a stronger correlation was found between these areas and the total CC impairment [18].

Furthermore, in 2018, Guduru, A. et Al., using OCTA demonstrated that in the para-atrophic hyperautofluorescent ring of STGD affected eyes a CC impairment is present. These authors argued that RPE thickening caused compression of the underlying choriocapillaris, with consequent ischaemic tissue damage [20]. 

As reported by de Carlo, et al., CC atrophy area was smaller than the corresponding regions of RPE and photoreceptor loss in STGD [21], suggesting a damage that starts from photoreceptors and then involves RPE and CC, all joined in a complex morpho-functional unit.

In our cohort of patients, CC damage, evaluated using OCTA, correlates with MP sensitivity in the extra atrophic areas, particularly in CC areas closer to the atrophy compared to more distant areas. 

Other authors showed how microperimetry is a reliable tool to assess retinal sensitivity in STGD patients with many cross-sectional and prospective studies [22,23]. Relationships between retinal sensitivity and retinal structure have been found with retinal sensitivity decreasing progressively at the advancing stage of the disease [24]; nevertheless, correlation with CC has not been investigated so far. It is predictable a decrease of both PPCA and retinal sensitivity proceeding from the center to the periphery and so a correlation between these parameters may be expected (in normal patients too), but not so far described. In our study we found this correlation to be stronger mainly in the pathologic areas around the atrophy in STGD.

In 2016 Bernstein, et al., have shown as fundus autofluorescence imaging does not describe adequately the central dense scotoma, and how MP can, with standardized grid testing, better define the border and allow more precise testing and longitudinal assessment of enlargement rates [19].

Limitations of our study include: (1) The small sample size, (2) the observational nature of the protocol, (3) the absence of a correlation with the autofluorescence and of a normal control group, and (4) the selection of patients with only advanced stages of STGD. A final limitation may be considered the fact that shadowing artifacts from hyperreflective material in the outer retinal layers may have confounded our analysis. However, we used a longer wavelength to image the CC, which significantly reduces shadowing artifacts and the deposits (“flecks”) are not usually of great dimension [25].

Further cross-sectional and longitudinal studies, with the analysis of the pre-atrophic stages of this pathology, may be considered useful to evaluate the disease progression and to better understand the pathogenesis.

In conclusion, according to our results, we demonstrated that in patients with STGD, close to the atrophic areas characterized by RPE and photoreceptor loss there is a dysfunctional RPE/photoreceptor complex that features retinal sensitivity impairment and probably CC impairment.

## Figures and Tables

**Figure 1 jcm-08-01432-f001:**
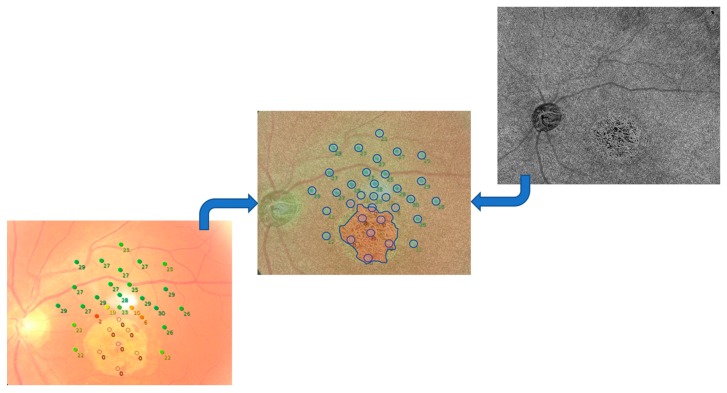
In the center retinography and MP grid overlapped on swept-source-optical coherence tomography angiography (SS-OCTA) choriocapillaris slab (in transparency) after a rigid transformation (OS of patient 1). In blue: circular areas centered on microperimetry (MP) points and the borders of the central atrophic area.

**Figure 2 jcm-08-01432-f002:**
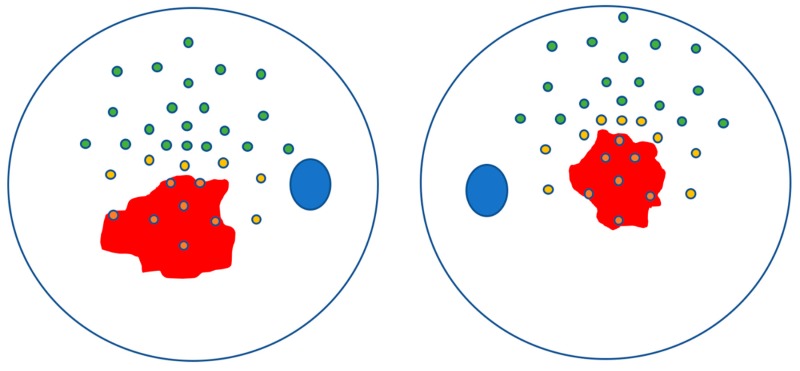
An example (outside atrophy (OA) of patient 1) of the sub-classification of the outside atrophy microperimetry points in near atrophy (NA) (in yellow) and distant from atrophy (DA) (in green). In red the atrophic area with the included points in orange.

**Figure 3 jcm-08-01432-f003:**
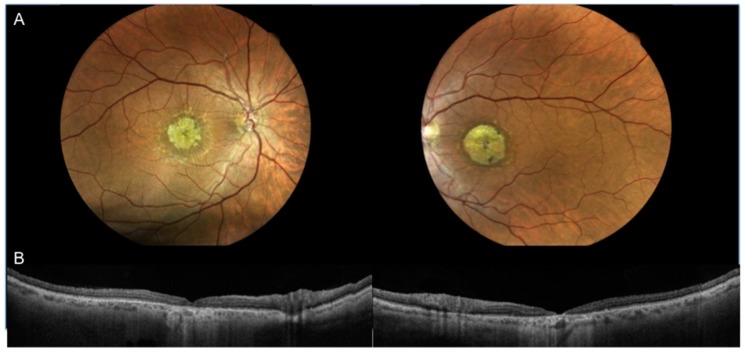
Case 2: Color fundus photographs (**A** right and left) showing macular atrophy with surrounding flecks and swept source optical coherence tomography angiography 12-mm scan (**B** right and left) passing through the fovea showing loss of inner and outer retinal layers ELM defect, inner segment ellipsoid band defect, thinner RPE/Bruchs complex, hyperreflective material located at the RPE level.

**Figure 4 jcm-08-01432-f004:**
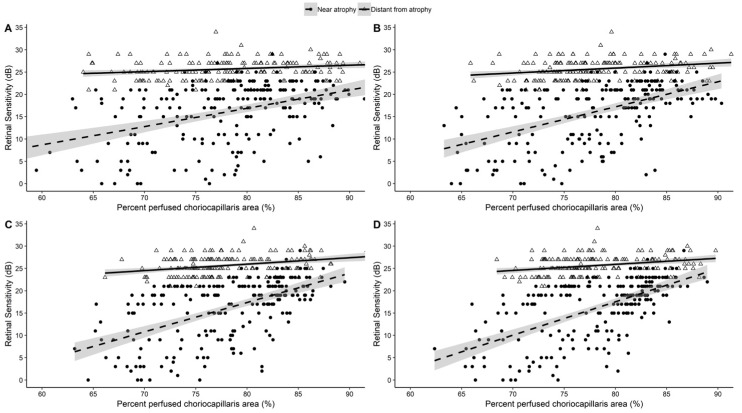
Scatter plots and linear regression lines showing the relationship between percent perfused choriocapillaris area and retinal sensitivity after stratification according to zone’s near atrophy and distant atrophy classification for each area considered. Panel (**A**) area 20 × 20 pixels, (**B**) area 40 × 40 pixels, (**C**) area 50 × 50 pixels, (**D**) area 60 × 60 pixels. Significant differences between linear regression coefficient were tested and all the *p*-values resulted <0.001.

**Table 1 jcm-08-01432-t001:** Clinical and genetic characteristics of patients in this study with Stargardt disease (STGD).

Patient	Sex	Age	VA	Fundoscopy	SD-OCT Features	CMT (µm)	Genotype (ABCA4 Mutations)
Case 1	Female	54 years	OD: 1.0	OU: Macular atrophy, flecks	OU: inner and outer retinal layer thinning/loss, ELM defect, ISe band defect, thinner RPE/ Bruchs complex, hyperreflective material located at the RPE level	OD 33	c.5882G > A p.(Gly1961Glu)
			OS: 1.0			OS 31	c.6089G > A p.(Arg2030Gln)
Case 2 *	Male	59 years	OD: l.2	OU: Macular atrophy, flecks	OU: inner and outer retinal layer thinning/loss, ELM defect, ISe band defect, thinner RPE/ Bruchs complex, hyperreflective material located at the RPE level	OD 70	c.4352+1A > G NA
			OS: 1.2			OS 65	c.5882G > A p.Gly1961Glu
Case 3 *	Male	56 years	OD: 0.8	OU: Macular atrophy flecks	OU: ELM defect, ISe band defect, photoreceptor outer segments defects,thinner RPE/ Bruchs complex, yperreflective material located at the RPE level	OD 120	c.4352 + 1A > G NA
			OS: 0.8			OS 125	c.5882G > A p.Gly1961Glu
Case 4	Female	55 years	OD: 1.0	OU: Macular atrophy, flecks	OU: inner and outer retinal layer thinning/loss, ELM defect, ISe band defect, thinner RPE/ Bruchs complex, hyperreflective material located at the RPE level	OD 32	c.6077T > C p.(Leu2026Pro)
			OS: 1.0			OS 45	c.4352þ1G > A p.(Ser1418_Pro1451delinsArg)
Case 5	Female	57 years	OD: 0.9	OU: Macular atrophy, flecks	OU: inner and outer retinal layer thinning/loss, ELM defect, ISe band defect, thinner RPE/ Bruchs complex, hyperreflective material located at the RPE level	OD 65	c.5882G > A p.(Gly1961Glu)
			OS: 1.1			OS 20	c.4234C > T p.(Gln1412 ^*^)
Case 6	Female	20 years	OD: 1.0	OU: Macular atrophy, flecks	OU: ELM defect, ISe band defect, photoreceptor outer segments defects, thinner RPE/ Bruchs complex, hyperreflective material located at the RPE level	OD 78	c.6112C > T p.(Arg2038Trp)
			OS: 1.0			OS 72	c.4462T > C p.(Cys1488Arg)
Case 7	Female	63 years	OD: 0.9	OU: Macular atrophy, flecks	OD: ELM defect, ISe band defect, photoreceptor outer segments defects, thinner RPE/ Bruchs complex, hyperreflective material located at the RPE level	OD 85	c.206G > A p.(Trp69 *)
			OS: NA		OS: NA	OS: NA	c.3113C > T p. (Ala1038Val)
Case 8	Male	47 years	OD: 1.2	OU: Macular atrophy	OU: inner and outer retinal layer thinning/loss, ELM defect, ISe band defect, thinner RPE/ Bruchs complex	OD 52	c.3322C>T p.(Arg1108Cys)
			OS: 1.2			OS 69	c.6112C>T p.(Arg2038Trp)
Case 9	Male	23 years	OD: 1.1	OU: Macular atrophy, flecks	OU: ELM defect, ISe band defect, photoreceptor outer segments defects,thinner RPE/ Bruchs complex, hyperreflective material located at the RPE level	OD 43	c.1622T>C p.(Leu541Pro)
			OS: 1.1			OS 59	c.6437G>A p.(Gly2146Asp)

VA, visual acuity; SD-OCT, Spectral Domain OCT; CMT, central macular thickness; RPE, retinal pigment epithelium; ELM, external limiting membrane; ISe, inner segment ellipsoid; NA, not available; * These two patients are brothers.

**Table 2 jcm-08-01432-t002:** Mean ± standard deviation of difference, respect to central atrophy values, of distant from atrophy and near atrophy zones for percent perfused choriocapillaris area (%) and retinal sensitivity (dB); p-values derived from linear mixed model between zones.

	Mean Differences ± SD		
Variable	Near Atrophy–Central Atrophy	Distant from Atrophy–Central Atrophy	*p-*Value
PPCA (%)			
Area 20 × 20 pixels	8.3 ± 4.3 *	12.6 ± 1.8 *	<0.001
Area 40 × 40 pixels	10.0 ± 5.5 *	14.6 ± 3.4 *	<0.001
Area 50 × 50 pixels	10.0 ± 5.2 *	15.0 ± 4.1 *	<0.001
Area 60 × 60 pixels	10.1 ± 4.9 *	15.5 ± 4.1 *	<0.001
Retinal Sensitivity (dB)	14.0 ± 4.9 *	25.7 ± 1.3 *	<0.001

* *p* < 0.001 after adjustment according to Bonferroni correction for multiple hypotheses testing vs. central atrophy. PPCA: Percent perfused choriocapillaris area.

**Table 3 jcm-08-01432-t003:** Coefficient derived from a linear mixed model regression between percent perfused choriocapillaris area at different area diameters (as independent variables) and retinal sensitivity in the overall population and after stratification in the near atrophy and distant from atrophy zones’ subgroups.

Area Diameters	Outside Atrophy		Near Atrophy		Distant From Atrophy	
	b ± SEM	*p-*Value	b ± SEM	*p-*Value	b ± SEM	*p-*Value
20 × 20 pixels	0.368 ± 0.050	<0.001	0.448 ±0.069	<0.001	0.040 ± 0.029	0.159
40 × 40 pixels	0.748 ± 0.059	<0.001	0.820 ±0.075	<0.001	0.122 ± 0.041	0.004
50 × 50 pixels	0.932 ± 0.062	<0.001	1.076 ±0.078	<0.001	0.166 ± 0.044	<0.001
60 × 60 pixels	1.196 ± 0.061	<0.001	1.502 ±0.067	<0.001	0.161 ± 0.048	0.001

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
