# Peer review of "Correlation between Choriocapillaris Density and Retinal Sensitivity in Stargardt Disease"

_jcm, 2019, doi:10.3390/jcm8091432_

Round 1

Reviewer 1 Report

No further comments. 

Reviewer 2 Report

The Authors have significantly improved the manuscript

This manuscript is a resubmission of an earlier submission. The following is a list of the peer review reports and author responses from that submission.

Round 1

Reviewer 1 Report

The Authors analyze the correlation with retinal sensitivity in STGD patients. The results indicate that CC damage on OCTA correlates with MP, suggesting that CC impairment may be a predictor of retinal function in patients with STGD. The study is interesting and original. I thaink that it deserves publication.

Reviewer 2 Report

The authors aim for a structure-function correlation in Stargardt's disease. It is an interesting approach usinng modern imaging and function testing devices. However the manuscript needs many improvements. Concerning methods, A central localisation of the MP grid is important for systematic comparisons between patients. PRL is different in each patient and temporal/sup/inf/nasal location could interfere with retinal sensitivity. The authors do not exclude possible shadowing by drusen in the NA and DA. Without doing so, you can not conclude on PPCA. The language needs improvements. Dealing with Stargardt, you would expect RPE atrophy and not CC atrophy as stated in some paragraphs. The main conclusion for CC impairment beeing a predictor can not be made as RPE atrophy is the main and primary finding in SD (as nown by pathogenesis). RPE atrophy features loss of sensitivity and likely CC atrophy. Here important refs are missing (Adhi M 2015, Muller PL 2017). So the findings might show a correlation but not any prediction as suggested for AMD (where pathogenesis is much more multifactoral).

Minor:

Figure 2 does not show the described areas

Which genetic testing was made/which genes investigated?

I do not think that you only included BCVA  >1.3 LogMar

How do you decide for qualtiy in MP testing? Without testrun, the values of MP are not reliabel!

How can an artifact occure in MP testing?

How was RPE atrophy defined?

Furhter Refs are missing at L 110

Reviewer 3 Report

Dear Authors,

I carefully reviewed the article entitled “Correlation Between Choriocapillaris Density and

Retinal Sensitivity in Stargardt Disease”.

The absence of retinal sensitivity measured by MP on atrophy is already well known.

It would be interesting to have the SD-OCT analysis as well. In fact, retinal sensitivity depends on the presence/absence of photoreceptors and RPE cells more than CC. The CC density might be an indirect marker for the integrity of photoreceptors/RPE cells. In this paper I cannot find any data regarding this point.

Conclusions: “this halo ….. represent a clinical useful tool to predict progression of atrophy”: I think this statement is not supported by results presented.